# DIVIDE AND CONQUER WITH NEURAL NETWORKS

**Alex Nowak**
Courant Institute of Mathematical Sciences
New York University
New York, NY 10012, USA
`anv273@nyu.edu`

**Joan Bruna** [*]
Courant Institute of Mathematical Sciences
New York University
New York, NY 10012, USA
`bruna@cims.nyu.edu`

## ABSTRACT

We consider the learning of algorithmic tasks by mere observation of input-output pairs. Rather than studying this as a black-box discrete regression problem with no assumption whatsoever on the input-output mapping, we concentrate on tasks that are amenable to the principle of *divide and conquer*, and study what are its implications in terms of learning.

This principle creates a powerful inductive bias that we exploit with neural architectures that are defined recursively, by learning two scale-invariant atomic operators: how to *split* a given input into two disjoint sets, and how to *merge* two partially solved tasks into a larger partial solution. The scale invariance creates parameter sharing across all stages of the architecture, and the dynamic design creates architectures whose complexity can be tuned in a differentiable manner.

As a result, our model is trained by backpropagation not only to minimize the errors at the output, but also to do so as efficiently as possible, by enforcing shallower computation graphs. Moreover, thanks to the scale invariance, the model can be trained only with only input/output pairs, removing the need to know oracle intermediate split and merge decisions. As it turns out, accuracy and complexity are not independent qualities, and we verify empirically that when the learnt complexity matches the underlying complexity of the task, this results in higher accuracy and better generalization in two paradigmatic problems: sorting and finding planar convex hulls.

## 1 INTRODUCTION

Many algorithmic tasks can be described as discrete input-output mappings, but this "black-box" vision hides all the fundamental questions that explain how and why the task can be optimally solved, which is the starting point of the study of algorithms and complexity. A powerful and general framework that breaks into this vision is the principle that many tasks have some degree of scale invariance or self-similarity, meaning that the ability to solve the task for a certain input size is essentially all that is needed in order to solve it for larger sizes. This principle is the basis of dynamic programming and is ubiquitous in most areas of discrete mathematics, from geometry to graph theory. In the case of images and audio signals, invariance principles are also critical for success: CNNs exploit both translation invariance and scale separation with multilayer, localized convolutional operators, which breaks the curse of dimensionality and brings the essential inductive bias explaining the success of CNNs. In our scenario of discrete algorithmic tasks, we build our model on the principle of *divide and conquer*, which provides us with a form of parameter sharing across scales akin to that of CNNs across space or RNNs across time.

While neural networks have been successful so far at providing flexible models for discrete regression and prediction tasks, mostly in Natural Language processing and discrete Reinforcement Learning, they are typically unaware and uninterested in complexity questions. Whereas some models are trained and tested at a fixed input/output scale (such as regression problems with generic fully connected neural networks), authors have explored ways to make training and testing less dependent of the input scale. The most prominent examples are Convolutional architectures, that

---

[*]Currently on leave from UC Berkeley

exploit translation invariance to accept variable size inputs by averaging their predictions at the last layer; and recurrent neural networks, that operate in an auto-regressive fashion to summarize any variable sized-input into a fixed-dimensional embedding. These two examples are paradigms of models whose complexity scales linearly with the input size.

Whereas CNN and RNN models define algorithms with linear complexity, attention mechanisms Bahdanau et al. (2014) generally correspond to quadratic complexity, with notable exceptions Andrychowicz & Kurach (2016). This can result in a mismatch between the intrinsic complexity required to solve a given task and the complexity that is given to the neural network to solve it. Our motivation is that learning cannot be 'complete' until these complexities match, and we start this quest by first focusing on problems for which the intrinsic complexity is well known and understood.

In this paper, we attempt to incorporate the complexity as yet another quantity that one wishes to minimize while training a model. We achieve this by using an architecture that learns recursively how to split a given input and learns how to merge each of the partial responses into a final output. Although these two steps could – and should – eventually be combined, in this work we start by exploring each of these architectures separately. We do so by only observing input-output pairs, getting away with the need to provide each of our artificial smaller problems with their correct output. This is a form of 'weak' supervision that is shown to work on tasks that are scale invariant, i.e. that can be addressed by divide and conquer. Another side benefit of our dynamic programming networks is their ability to generalize to larger scales. By construction, our model learns the same decision at each scale, and therefore can generalize well whenever the task is compatible with that inductive bias.

**Summary of Contributions:**

- We introduce a recursive split and merge architecture, and a learning framework that optimizes it not only for accuracy but also for computational complexity in a fully differentiable manner, using only input-output example pairs.

- We provide preliminary empirical evidence that the dynamic programming principle can be efficiently learnt on simple tasks such as sorting and planar convex hull.

## 2 RELATED WORK

Using neural networks to solve algorithmic tasks is an active area of current research, but its models can be traced back to context free grammars Fanty (1994). In particular, dynamic learning appears in works such as Pollack (1991) and Tabor (2000).

The current research in the area is dominated by Recurrent Neural Networks Joulin & Mikolov (2015); Grefenstette et al. (2015), LSTMs Hochreiter & Schmidhuber (1997), sequence-to-sequence neural models Sutskever et al. (2014); Zaremba & Sutskever (2014), attention mechanisms Vinyals et al. (2015b); Andrychowicz & Kurach (2016) and explicit external memory models Weston et al. (2014); Sukhbaatar et al. (2015); Graves et al. (2014); Zaremba & Sutskever (2015). We refer the reader to Joulin & Mikolov (2015) and references therein for a more exhaustive and detailed account of related work.

Amongst these works, we highlight some that are particularly relevant to us. Neural GPU Kaiser & Sutskever (2015) defines a neural architecture that acts convolutionally with respect to the input and is applied iteratively $o(n)$ times, where $n$ is the input size. It leads to fixed computational machines with total $o(n^2)$ complexity. Neural Programmer-Interpreters Reed & de Freitas (2015) introduce a compositional model based on a LSTM that can learn generic programs. It is trained with full supervision using execution traces. Hierarchical attention mechanisms have been explored in Andrychowicz & Kurach (2016). They improve the complexity of the model from $o(n^2)$ of traditional attention to $o(n \log n)$, similarly as our models, but they are trained very differently, using REINFORCE. Finally, Pointer Networks Vinyals et al. (2015b;a) modify classic attention mechanisms to make them amenable to adapt to variable input-dependent outputs, and illustrate the resulting models on geometric algorithmic tasks. It belongs to the $o(n^2)$ category class.

## 3 DIVIDE AND CONQUER WITH NEURAL NETWORKS

In this section we present our basic model architecture with its core Split and Merge blocks, and then describe how to build the global dynamic programming network. In our formulation, we choose generic data structures for split and merge. The split module accepts sets of elements as input, and outputs a disjoint partition of this set. The merge module accepts two ordered sets $\Omega_1$ and $\Omega_2$ as input, and produces a subset $\Lambda \subseteq \Omega_1 \cup \Omega_2$ of their union that respects the partial ordering; i.e, if $x_1 \succ_{\Omega_1} x_2$, $x_1, x_2 \in \Omega_1 \cap \Lambda$, then necessarily $x_1 \succ_\Lambda x_2$.

### 3.1 SPLIT

Split blocks receive an input set of elements $\Omega$, and output a partition $\Omega = \Omega_1 \cup \Omega_2$. The corresponding neural network architecture is permutation invariant. We use a simple block that computes nonlinear moments of the input. Denote $n = |\Omega|$ and assume the elements of $\Omega$ to be in $\mathbb{R}^d$. Define the matrix $x \in \mathbb{R}^{d \times n}$ as the elements in $\Omega$ organized in columns under an arbitrary ordering and $x_i$ as the $i$-th column. This architecture must take sets as inputs, i.e, $(S(x, \theta))_\sigma = S(x_\sigma, \theta) \, \forall \sigma \in \mathfrak{S}_n$ (where $\sigma$ acts in rows). We propose a $\Theta(n)$ architecture in Appendix A, a somewhat simpler version of similar set-to-set models such as those in Vinyals et al. (2015a); Sukhbaatar et al. (2016). We can compute both subsets either by sampling from the output probabilities or taking the mode ($\mathbb{1}[p_i \geq 0.5]$).

### 3.2 MERGE

Merge blocks receive two ordered inputs and produce a subset of their union that preserves the partial ordering. We can visualize the merge block as having two different tasks:

- Choose a mask over each input to rule out some elements.
- Merge both subsets preserving the partial ordering.

In this paper, for simplicity, we only learn the first task of the merge block and leave the learnability of the second for future work. Let $y_1$ and $y_2$ be two ordered sequences of elements in $\mathbb{R}^d$ of lengths $n_1$ and $n_2$ respectively. We parametrize the first part of the merge block using Bi-directional LSTMs. The input of the merge block will be a sequence of vectors in $\mathbb{R}^{d+1}$. The first $d$ dimensions of the elements in the sequence will correspond to the concatenation of $y_1$ and $y_2$. The last dimension of each element in the input sequence will be a boolean indicating to which sequence the element belongs. We produce the element-wise probabilities by concatenating the hidden states at every element and computing $p_i = \sigma(w^T[h_1, h_2] + b)$ for $i = 1, \dots, n_1$ and $p_i = \sigma(w^T[h_2, h_1] + b)$ for $i = n_1 + 1, \dots, n_2$ where $b$ is a scalar bias and $w$ is a vector in $\mathbb{R}^{2n_h}$ with $n_h$ being the number of hidden units of each LSTM. We can also compute both subsets either by sampling from these probabilities or taking the mode ($\mathbb{1}[p_i \geq 0.5]$).

To merge the chosen subsets we will use a deterministic procedure which will produce the correct output if both subsets are correct and will be stable for small errors in the mask. This procedure will be specific for each task and we explain our choice in the experiments section when addressing the convex hull task.

### 3.3 BUILDING THE MODEL

Figure 1 illustrates our architecture. The divide and conquer principle is implemented by successively splitting inputs until the input $\mathbf{x}_0$ of size $|\mathbf{x}_0| = n$ is broken into a collection of sets $\{\mathbf{x}_v\}_{v \in V_i}$, $i \leq l$ such that they reach a critical size $K_0$. At that critical scale we consider that the task at hand can be solved with a generic mapping with constant complexity to produce the outputs $\{\mathbf{x}_v\}_{v \in V_i}$. We consider problems where the outputs are defined as partially ordered subsets of the input. The second phase consists in merging this collection of outputs into the global solution. As described earlier, in this work we consider the two phases separately, and leave the joint training of split and merge operations for future work.

The model has parameters for each split and merge, but these parameters are shared across all the instances. Also, the structure of the binary tree is dynamic: each input determines the respective sizes of the split, which in turn determine the length of each corresponding branch of the tree.

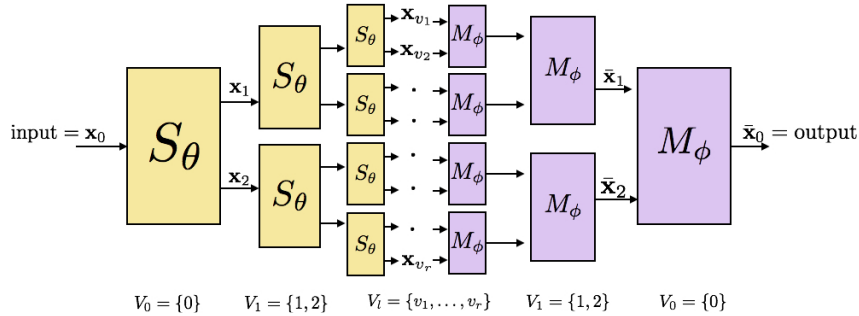

Figure 1: Instantiation of our Dynamic Programming Net. An input instance is recurrently split into smaller instances until we reach a critical scale. The merge procedure works in the opposite direction. We highlight the fact that split and merge sizes can vary and are data dependent, and therefore the binary tree is sample-dependent. All split blocks share the same parameters $\theta$ and similarly for the merge blocks with $\phi$.

## 4 LEARNING

The training objective is to both maximize the accuracy at the output, but also to be able to do so with minimal complexity. This section describes how our dynamic architecture allows to optimize both objectives with gradient descent.

### 4.1 ADJUSTING COMPLEXITY WITH GRADIENT DESCENT

The average case complexity of the split phase satisfies the following recursion:

$$\mathbb{E}C_s(n) = \mathbb{E}\{C_s(\alpha_s n) + C_s((1 - \alpha_s)n)\} + A \cdot n \,, \tag{1}$$

where $(\alpha_s, 1 - \alpha_s)$ are the fraction of input elements that are respectively sent to each output and $A \cdot n$ is the cost of running our split module described in Section 3.1. Since this fraction is input-dependent, the average case is obtained by taking expectations with respect to the underlying input distribution. Assuming without loss of generality that $\mathbb{E}(\alpha_s) \geq 0.5$, the resulting complexity is of the order of

$$\mathbb{E}C_s(n) \simeq \frac{An \log n}{\log \mathbb{E}\alpha_s^{-1}} \,. \tag{2}$$

The overall complexity of the resulting model can be thus partially controlled by enforcing $\alpha_s$ to be as close as possible to 0.5. The range $\alpha_s \in [0.5, 1)$ gives us the ability to span complexities between $\Theta(n \log_2 n)$ (perfectly balanced trees) and $\Theta(n^2)$ (perfectly unbalanced trees with $n - 1$ and 1 elements).

Similarly, our merge selection phase performs a selection of a subset of the union of its two inputs. Let $n = |\Omega_1| + |\Omega_2|$ be the total amount of incoming data at any given node and $\alpha_m = \frac{|\Lambda|}{n}$ denote the fraction of elements that is sent to the next level. If we assume an execution tree given by a split phase with factor $\alpha_s$, we verify that the resulting complexity satisfies

$$C_M(n) = Bn\alpha_m^{\frac{-\log n}{\log \alpha_s}} + C_M(n\alpha_s) + C_M(n(1 - \alpha_s)) \,. \tag{3}$$

When $\mathbb{E}\alpha_m < 1$, by applying the Akra-Bazzi method Akra & Bazzi (1998) this results in

$$\mathbb{E}C_M(n) = \Theta(n) \,, \tag{4}$$

and $\mathbb{E}C_M(n) = \Theta(n \log n)$ when $\mathbb{E}\alpha_m = 1$. The merge block thus operates in linear time as soon as each merge operation removes a constant fraction of its inputs at each scale. We shall impose that $\mathbb{E}\alpha_m \geq \mathbb{E}\alpha_s$ to ensure that the merge decisions do not reach the target size before the tree has reached its end. This creates an incentive to make decisions to discard elements early in the process, but as we shall see in the next section, this is offset by the fact that labeled information exists only at the bottom of the merge process. In the next subsections we shall see how to control the branching factors in a differentiable fashion, enabling learning by simple gradient descent.

## 4.2 WEAK VERSUS STRONG SUPERVISION

A key requirement of our approach is the ability to learn only from input-output pairs of training examples. Since the examples have arbitrary size, in general we have no supervision for each individual instance of split and merge operations. The challenge is that each of these blocks sends information to the next by making discrete sampling decisions.

One possibility is to embark in optimization strategies that differentiate under sampling, such as those arising in Reinforcement Learning and in particular the REINFORCE. However, these optimization methods involve gradient estimates with large variance, resulting in poor sample complexity. We study an alternative that exploits the powerful inductive bias given by sharing parameters at all scales and also the fact that we optimize for both accuracy and complexity, and requires only backpropagation and no intermediate sampling.

We first focus on the training of each split and merge phase separately, and we assume first that we have available labels at the output of each of these phases. That is, we assume we have a dataset $\{(\mathbf{x}^k, \mathbf{y}^k)\}_{k \leq K}$ of $K$ examples. In the case of the split phase, we assume that $\mathbf{x}^k$ is a set of $n_k$ (possibly varying with $k$) elements, and $\mathbf{y}^k$ is an ordered partition of $\mathbf{x}^k$. In the case of the merge, $\mathbf{x}^k$ represents an ordered partition of a certain set $\mathbf{z}_k$, and $\mathbf{y}^k$ is a certain (ordered) subset of $\mathbf{z}_k$. Each of these subproblems already contains the challenge of training several instances of split and merge connected through discrete, non-differentiable sampling operations. We shall now describe in detail our strategy to compute gradients with respect to parameters $\theta$ (split) and $\phi$ (merge) just with backpropagation.

## 4.3 TRAINING SPLIT

In order to train the split block we must create artificial targets at every node of the generated tree from the available final target partition, which is an ordered sequence of disjoint subsets of the input.

Figure 1 shows that every scale of the generated tree corresponds to a partition of the final target, and every node of the tree at that scale corresponds to one of the sets of the partition, whose size equals the length of the input at that node.

For simplicity, we drop here the superscript denoting the training instance. Let us denote by $\Omega_v = \mathbf{x}_v \cap \mathbf{y}_v$ the intersection of the input $\mathbf{x}_v$ at node $v$, and the corresponding target subset $\mathbf{y}_v$. The elements in that intersection will provide gradient signal to update $\theta$ for each node as follows. The split block at $v$ with current parameters computes the vector of probabilities $p_v(\theta, \mathbf{x})$, encoding the probability that each element in $\mathbf{x}_v$ will be sent towards the first output or the second. In order to create targets for these outputs, we first sample from $p_v(\theta, \mathbf{x})$ to obtain a partition of $\mathbf{x}_v$: $\mathbf{x}_v = T^1 \cup T^2$. This partition in turn defines a 'valid' partition $\Omega_v = \Omega_v^{\mathrm{inp},1} \cup \Omega_v^{\mathrm{inp},2}$, with $\Omega_v^{\mathrm{inp},j} = \Omega_v \cap T^j$ for $j = 1, 2$.

Similarly, we consider the target partition of the same size defined by the order in the target subset $\Omega_v = \Omega_v^{\mathrm{targ},1} \cup \Omega_v^{\mathrm{targ},2}$ where $|\Omega_v^{\mathrm{targ},i}| = |\Omega_v^{\mathrm{inp},i}|$, $i = 1, 2$. This partition creates the targets

$$t = \mathbf{1}(x \in \Omega_v^{\mathrm{targ},1}) , \ x \in \Omega_v .$$

As described in Section 4.1, besides providing targets corresponding to correct solutions, we also attempt to minimize the average complexity of the model. The contribution of node $v$ to the loss is thus

$$l(\theta, \mathbf{x}_v, \mathbf{y}_v) = - \sum_{x \in \Omega_v^{\mathrm{targ},1}} \log(p_v(\theta, x)) - \sum_{x \in \Omega_v^{\mathrm{targ},2}} \log(1 - p_v(\theta, x)) - \beta_S R(v, \theta) , \quad (5)$$

where

$$R(v, \theta) = \frac{1}{|\Omega_v|} \left( \sum_{x \in \Omega_v} p_v(\theta, x)^2 \right) - \left( \frac{1}{|\Omega_v|} \sum_{x \in \Omega_v} p_v(\theta, x) \right)^2 \quad (6)$$

is a regularization term similar to an empirical variance which will encourage the split block to partition the input into equal parts when maximized. Increasing $\beta_S$ will give more preference to split inputs into equal parts, but using a large value can make the performance go down.

We can finally define the total loss by aggregating the losses across all the nodes of the tree:

$$L^S(\theta, \eta, \mathbf{x}, \mathbf{y}) = \sum_{i=0}^{l-1} \eta_i \frac{1}{|V_i|} \sum_{v \in V_i} l(\theta, \mathbf{x}_v, \mathbf{y}_v) \,, \tag{7}$$

where $V_i$ is the set of vertices at depth $i$, and $\eta = (\eta_1, \ldots, \eta_l)$ is a vector of dynamic hyperparameters that can be changed during training and $\beta_S$ is a real positive hyperparameter.

REMARK: if $\Omega_v = \emptyset$, the corresponding node will not be trained because we can't create a target for it.

We will consider $\eta$ as being a binary vector, but non-binary $\eta$ can make sense in some situations. Its role is to dynamically control which scales are trained and which are not, so putting $\eta_i = 1$ will make the $i$-th scale trainable. Observe that there is a hierarchy in the generated tree, i.e, the ability to discriminate at a given node will strongly depend on the performance at smaller depths. This ability can be quantified by $|\Omega_v|$, which will increase as performance of previous nodes becomes better. This observation encourages us to give more preference at the top nodes at the beginning of the training, which motivates the definition of the dynamic hyperparameter $\eta$, and is a form of automatic curriculum learning, since by construction the model has to first learn to do well at the coarsest scales before targets can be defined at the finer scales.

REMARK: If we just train the split architecture neglecting the merge, there is no need to first split recursively the input storing the activations at every node and train all the nodes afterwards. We can train the nodes while we are generating the tree. This training procedure reduces considerably the amount of storing space needed for every batch. However, our goal in future work is to train both architectures together. In this case, we must save the activations because the final target for the split will only be available after a full forward pass through split and merge.

## 4.4 TRAINING MERGE

We describe here the procedure to learn how to perform the merging selection described in Section 3.2. Analogously to the split, the key point on the merge training is how to build proper targets at every node of the tree having only the final target available $\mathbf{y}$. In this case, this question is more complicated because the merge block not only merges the two inputs preserving the inner ordering of each, but also rules some of them out. Since we only observe input-output pairs, input elements that are not in the target should be ruled out, but we don't know at which scale of the tree they should be discarded. Let us describe how again thanks to the scale invariance and with appropriate regularization, we can train these operation with only external supervision.

The input $\mathbf{x}$ of the merge architecture will be an ordered sequence of disjoint subsets. They correspond to an underlying split tree, so they can be seen as the leaves of a binary tree. First we forward the input to the architecture and merge the outputs of each block recursively storing the activations at every node until we reach the root node. To create the output mask from the probabilities at every node, we always pick the elements belonging to the final convex hull, and the rest will be sampled according to the probabilities. This way, the target elements will always appear in one node at every scale. Denote by $\mathbf{x}_v^1, \mathbf{x}_v^2$ the inputs at node $v$ and $\mathbf{y}_v = \mathbf{y} \cap (\mathbf{x}_v^1 \cup \mathbf{x}_v^2)$ the intersection of both inputs with the final target. Let $p_v(\phi, x_i, \mathbf{x}_v^{\{1,2\}})$ denote the probability computed by the merge model that element $x_i$ is discarded at node $v$. The contribution of node $v$ to the total loss is defined as

$$l(\phi, \mathbf{x}_v^1, \mathbf{x}_v^2, \mathbf{y}_v) + \beta_M R(\alpha_M, \mathbf{x}_v^1, \mathbf{x}_v^2) = - \sum_{x_i \in \mathbf{y}_v} \log p_v(\phi, x_i, \mathbf{x}_v^{\{1,2\}}) + \beta_M \left( \sum_{x_i \in \mathbf{x}_v^1, \mathbf{x}_v^2} p_v(\phi, x_i, \mathbf{x}_v^{\{1,2\}}) - \alpha_M |\mathbf{x}_v^1 \cup \mathbf{x}_v^2| \right)^2 . \tag{8}$$

 Here, $\beta_M$ is the hyperparameter controlling the tradeoff between accuracy and complexity, and $\alpha_M$ is a shrinkage factor that models the rate by which each input is shrunk at each scale. This shrinkage rate depends on the task at hand and we have not enough supervision in our setup to estimate it from the data. We thus settle for a rate that is scale invariant, that is the total amount of elements at scale $i$ will follow a law of the form $\alpha_M^i$. Notice that the cross-entropy term is unbalanced: it only penalizes false negatives, since false positives can be recovered at other scales, but not false negatives.

Finally, we define the total loss of the architecture as

$$L^M(\phi, \mathbf{x}, \mathbf{y}) = l_0(\phi, \mathbf{x}_{v_0}^1, \mathbf{x}_{v_0}^2, \mathbf{y}_{v_0}) + \sum_{i=1}^{l-1} \sum_{v \in V_i} \left[ l(\phi, \mathbf{x}_v^1, \mathbf{x}_v^2, \mathbf{y}_v) + \beta_M R(\alpha, \mathbf{x}_v^1, \mathbf{x}_v^2) \right] \text{ , with} \quad (9)$$

$$l_0(\phi, \mathbf{x}_v^1, \mathbf{x}_v^2, \mathbf{y}_v) = - \sum_{x_i \in \mathbf{y}_v} \log p_v(\phi, x_i, \mathbf{x}_v^{\{1,2\}}) - \sum_{x_i \in (\mathbf{x}_v^1 \cup \mathbf{x}_v^2) - \mathbf{y}_v} \log(1 - p_v(\phi, x_i), \mathbf{x}_v^{\{1,2\}}) . \quad (10)$$

On the last node of the tree we can penalize for both false positives and negatives because its node target corresponds to the final target.

## 5 EXPERIMENTS

Experiments are implemented in Tensorflow, with reproducible code soon available at `https://github.com/alexnowakvila/DP`.

### 5.1 SORTING

We implemented the split architecture for the task of sorting. This is a good task to test the split block because we can solve it using an oracle splitting block (*quicksort*), which consists in finding a centered pivot – such as the median. The final targets will be the sorted vector and the input will be the set of elements of the vector. In this case, the dimensionality of the input is $d = 1$ because we are sorting scalars.

We train the model for vectors of length $n = 256$ and train until depth 8. We use 40 layers for the sorting block defined in Appendix A, resulting in a total of 240 parameters for the architecture in a whole. The dataset has 4096 input-output pairs and we train during 5 epochs, by varying the input distribution; see Figure 2. The training is performed using batches of size 32. At the beginning we only train the first two scales and train one step deeper every 100 batches.

As described in Appendix A, we normalize the input set before feeding it into the next block using the empirical mean and standard deviation. This normalization is lossless if one also feeds these two empirical moments to the split block, as explained in Appendix A, but we observed no noticeable change in the performance in the sorting case. This is consistent with the fact that sorting is invariant to affine transformations of the input. Similarly as in Ioffe & Szegedy (2015), normalizing the input by its first two moments reduces the effect of numerical instabilities and reduces covariate shift.

We sample from the output probabilities during training to connect different blocks. Sampling is shown to be helpful during the first steps of training because it provides some exploration and avoids the probabilities to get stuck. However, we observed an increase in performance using the mode at test time. We measure our accuracy using the ratio between the inversions of the output and the mean number of inversions. We denote it by Inversions Ratio (IR): $IR = \frac{\text{inversions}}{\frac{1}{2}\binom{n}{2}}$. We trained the model using normal and exponential distributions. If we use a uniform distribution the splitting task reduces to split by the mean and this can be achieved with a much simpler block. The results show an impressive generalization performance with input length and robustness with respect to input distribution. It is remarkable to achieve this only using weak supervision (i.e, input-output pairs). Figure 2 presents our numerical results and its analysis.

### 5.2 PLANAR CONVEX HULL

We trained the merge architecture for the planar convex hull with both weak and strong supervision assuming an oracle split, i.e, the inputs are disjoint convex hulls. By now, for simplicity, the model only learns how to choose a subset of each of both inputs, but not how to merge both subsets. The deterministic policy to merge both chosen subsets is the following. The procedure will compute angles between the vector that goes from the barycenter to every point and the unit vector $(-1, 0)$. The merged ordered sequence will be the points sorted by the corresponding angles. This policy will produce outputs which are stable for small errors in the masking. We used 15 hidden units for each LSTM, giving a total number of parameters of 2511. We trained with batches of size 16 over a training set of size 1024.

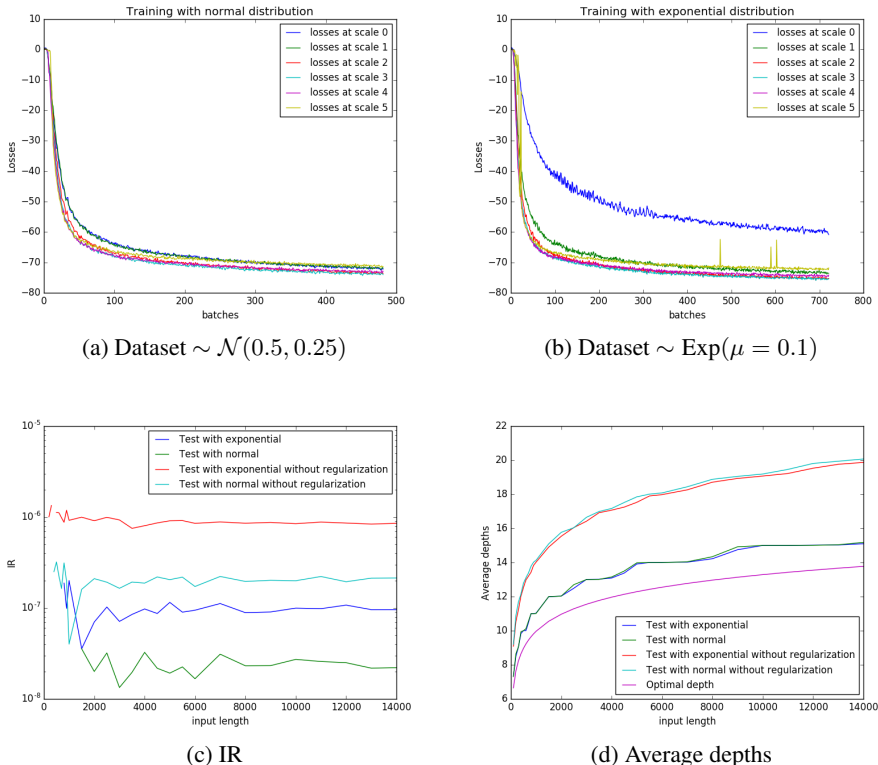

(a) Dataset $\sim \mathcal{N}(0.5, 0.25)$

(b) Dataset $\sim \mathrm{Exp}(\mu = 0.1)$

(c) IR

(d) Average depths

Figure 2: (a) and (b): The loss at every scale is computed averaging over all the losses of the nodes for all elements in the batch. We only show the first 6 scales. The input distribution for every node of the generated tree is different at every block and scale because the inputs at a given scale are the result of a finer partition of the inputs at the previous one. (a): *Losses during training for data set* $\sim \mathcal{N}(0.5, 0.25)$. We observe that the values of the losses are very similar. However, the losses at first scales are a little bit larger than the others because the input distribution becomes more symmetric as the partition gets finer. (b): *Losses during training for data set* $\sim \mathrm{Exp}(\mu = 0.1)$. When training using the exponential distribution with mean $\mu = 0.1$ as input, we observe that due to it's high non-symmetry, the losses at the first scale have a larger value. However, the difference between losses shrinks during training. (c) and (d): The following tests are performed with the architecture trained using the normal distribution and all the points are computed averaging over 100 examples. (c): *Inversions Ratio (IR)*. The generalization test results are impressive. We test from $n = 100$ to $n = 14000$ and the accuracy converges to a value of magnitude $10^{-6} - 10^{-7}$. If there is no curve at the beginning of the test means that the number of inversions is 0. The results show that the model performs better when the input distribution is normal because the resulting input distributions at every block are easier to split into equal parts. Without regularization the results are worse. (d): *Average depths*. We plot the average depths of the generated tree. The depth is logarithmic with respect to the size of the input. The curve is independent of the test distribution and the regularization term is key to make the complexity of the algorithm be close to the optimal.

It is important to prevent the model to learn the resulting convex hull just by the absolute position of the points. For instance, if the points were drawn from a unit distribution on the unit square, then the model could guess if a point belongs to the convex hull just using its absolute position. As the loss in the weak supervision framework only discriminates between points belonging to the final convex hull, using the absolute position can lead to a bad performance in the intermediate scales. In order to get around this problem, our points were drawn from a uniform distribution on a half unit square $[0.5, 0.5]^2$ whose center is a random point in the square $[0.25, 0.75]^2$. This way, we reduce considerably the reliability of the absolute position to solve the task.

We first train the model using strong supervision, i.e, every node of the tree will use the correct target and all inputs at all nodes will be the correct convex hulls. We can think of strong supervision as a sort of curriculum learning, but here, the block is simultaneously learning with different lengths. At test time, we create the output mask by taking the mode of each output probability (can also try with probabilities). To train using weak supervision, we must put a prior for the complexity of the whole algorithm using $\alpha_M$. Its optimal value is unknown if just using input-output pairs, and in fact, it depends on the scale and the data distribution. However, we estimate it by $\alpha_M = \left( \frac{|\Omega_t|}{\sum_i |\Omega_i|} \right)^{1/l}$ where $\Omega_t$ is the target, $\Omega_i$ is the $i$-th input and $l$ is the number of scales.

At test time, we measure the accuracy of the model with the following quantity: Accuracy $= \frac{|\Omega_t \cap \hat{\Omega}|}{|\Omega_t \cup \hat{\Omega}|}$ . We adjusted the hyperparameter $\beta_M$ by making the gradient norms of the first part of the loss 8 to be of the same magnitude of the gradient norms of the regularization term. We found the optimal $\beta_M$ to be $10^{-2}$. We observed a much worse convergence using a $\beta_M$ larger than the optimal in the case of 3 scales. We also confirmed the necessity of the regularization term for the model to converge (see Figure 3). The average training time per epoch depends on the total number of scales and the experimental $\alpha_M$. For instance, the model trained using WS takes about 1:30 min and 2 min in average for 2 and 3 scales respectively. However, when training using WS for 3 scales without regularization term, the probabilities get stuck at 1, producing an increase in complexity of the model and a training time per epoch of 6 min.

## 6 DISCUSSION

We have presented a framework that has the ability to leverage an important smoothness prior present in many discrete algorithmic tasks, namely the scale invariance or the ability to divide and conquer. Similarly as the local translation invariance prior when learning with images using CNNs, exploiting this inductive bias breaks the curse of dimensionality and provides a solid foundation to learn complex functional dependencies.

Our approach is an attempt to mimic the behavior of dynamic programming algorithms with neural networks. It is instantiated with two atomic operations – split an input into two, and merge two outputs into a single one – that are recursively applied. This framework allows us to train the system using weak supervision (that is, by only observing input-output pairs), and gives us another bullet: the ability to optimize not only for accuracy but also for complexity, all in a fully differentiable setup. Our numerical results are by all means preliminary, and much is still to be done before this architecture becomes competitive. In particular, we are considering the following directions.

*Current work:* We are currently working in two major aspects. The first one is to generalize the merge step so that it can not only perform the selection but also the concatenation in a fully learnt manner. This operation can also be implemented with $o(n)$ architectures since the partial order at the output respects the partial order in each of the two inputs. The second one is to perform the joint training of both split and merge blocks, which will allow us to learn with *really* weak supervision. For that purpose, it is necessary to perform target propagation to provide suitable targets for the split block. Lastly, we are in the process of comparing our results with standard baselines that do not exploit the dynamic structure of the problem.

*Future work:* The number of extensions and applicability of this model is vast. In particular, we want to extend split and merge architectures in graph problems such as shortest paths or spanning trees. For that purpose, we will use Graph Neural Networks Scarselli et al. (2009); Sukhbaatar et al. (2016) as atomic models to handle the data. We also want to explore more generic architectures for the atomic split and merge, perhaps with $\Theta(n \log n)$ complexity, to cover more territory between linear and quadratic complexity, and to test the ability of the model to learn good approximations in tasks that are NP-hard. Another question that this model raises is the consistency of weak supervision thanks to the scale invariance. A byproduct of our scale invariance seems to be the ability to propagate and diffuse the available targets at the coarsest scale at all scales, leading to 'self-consistent' supervision. We believe this is a profound question. Finally, we want to explore the links between these ideas and that of the hierarchical Reinforcement learning, which is an extreme form of very weak supervision. Designing intermediate rewards is akin to our setup of defining suitable targets at intermediate scales, albeit with an extra degree of difficulty.

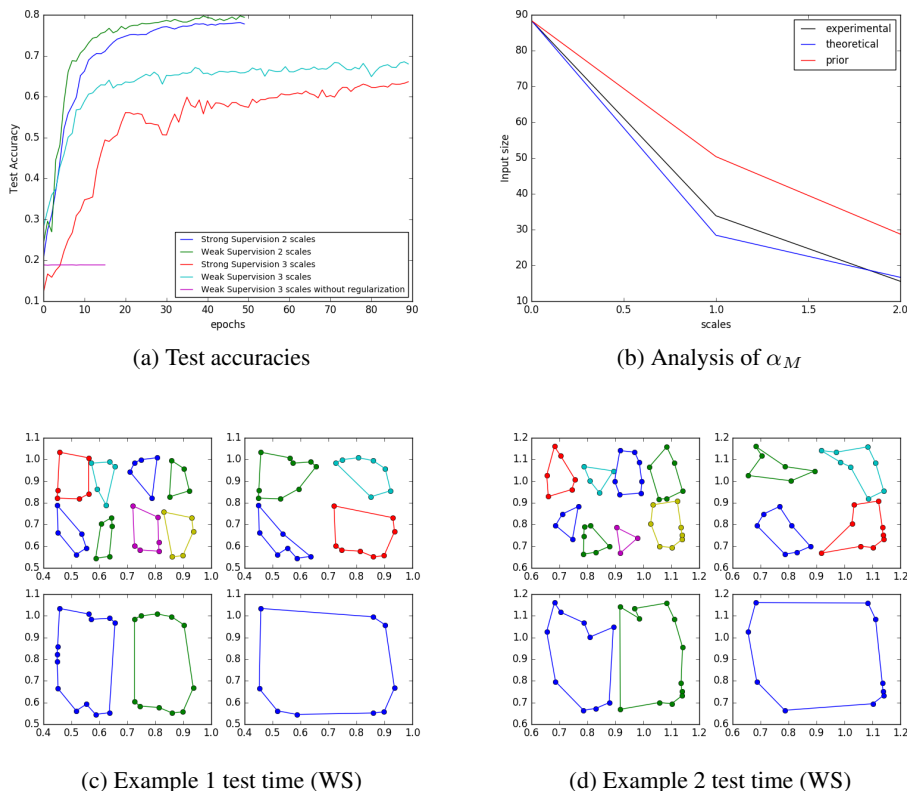

(a) Test accuracies

(b) Analysis of $\alpha_M$

(c) Example 1 test time (WS)

(d) Example 2 test time (WS)

Figure 3: After every epoch, we test the model in a dataset of size 128. (a): *Test accuracies*. We observe that the accuracy is lower for 3 scales. This is due to the tree structure and how the errors propagate through it. In the case of SS, the merge block is not able to correct mistakes coming from previous layers because the model is always trained with correct inputs. For WS, we also observe a lower accuracy for 3 scales, however, the decrease is smaller than using strong supervision. We claim that the reason for this is that the block is able to correct mistakes from previous layers because the inputs of the blocks during training are not perfect convex hulls. We also show that without regularization the model is not able to learn with WS. (b): *Analysis of $\alpha_M$*. We plot the average number of masked elements at every scale over a batch of size 16. The theoretical masks are computed using the correct dynamic algorithm, the estimate masks are the ones computed using the prior $\alpha_M$ used during training and the experimental corresponds to the masks when doing a forward pass using the trained model. (c) and (d): *Examples at test time (WS)*. In these two examples we show how the model is able to correct mistakes at previous scales and producing the output convex hull with good accuracy.

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

# A   DETAILS ON SPLIT ARCHITECTURE

## A.1   SPLIT

Let's define for $k = 1, \ldots, r$ layers

$$F_i^0 = x_i, \qquad F_i^k = F_i^k(x_i, z^{k-1}, F_i^{k-1}, \theta) \in \mathbb{R}^d \qquad i = 1, \ldots, n$$

where $z^k = \frac{1}{n} \sum_{i=1}^n F_i^k$ is the empirical moment defined by $F_i$ and $\theta$ is a set of parameters. Then, $p(x) = \sigma(F^r)$ is the vector of probabilities, i.e, $p_i = \mathbb{P}(x_i \in \Omega^1)$. This is the most general expression of our architecture. The idea is that the hidden units in layer $k$ at position $i$ depend on $x_i$ (input at same index), $z^{k-1}$ (average of all hidden states of previous layer) and $F_i^k$ (previous hidden unit at same index).

We reduce the covariate shift of the split architecture by normalizing the input sets by their empirical mean and covariance:

$$\tilde{x} = \frac{x - \hat{\mu}}{\hat{\sigma}} \ .$$

Since both $\hat{\mu}$ and $\hat{\sigma}$ are two empirical averages as the $z^k$'s, they can be seamlessly integrated as extra averages by concatenating them to $z^k$.

We have parametrized the functions $F_i^k$ in the following way:

$$F_i^k = \phi_0^k \times F_i^{k-1} + \phi_1^k \times \phi_2^k$$

where $\phi_0^k, \phi_1^k$ and $\phi_2^k$ have the following form:

$$\phi_0^k = \sigma(C_0^k x_i + W_0^k z^{(k-1)} + b_0^k) \,, \ \phi_1^k = \sigma(C_1^k x_i + W_1^k z^{(k-1)} + b_1^k) \,, \ \phi_2^k = \tanh(C_2^k x_i + W_2^k z^{(k-1)} + b_2^k) \,,$$

and $C_i^k, W_i^k$ are $d \times d$ matrices and $b_i^k \in \mathbb{R}^d$ a bias vector. Note that the total number of parameters is $r(3d^2 + 3d)$ where $r$ is the number of layers and $d$ is the dimensionality of the input elements.

