# Peer review of "Divide and Conquer with Neural Networks"

_ICLR 2017 — rejected_

[Public Comment · Tara N Sainath · 07 Nov 2016]
**ICLR Paper Format**

Dear Authors,

Please resubmit your paper in the ICLR 2017 format with the correct margin spacing for your submission to be considered. Thank you!

[Official Review · AnonReviewer2 · rating 3 · confidence 2 · 15 Dec 2016]
**Interesting but extremely difficult read**

I find this paper extremely hard to read. The main promise of the paper is to train models for combinatorial search procedures, especially for dynamic programming to learn where to split and merge. The present methodology is supposed to make use of some form of scale invariance property which is scarcely motivated for most problems this approach should be relevant for. However, the general research direction is fruitful and important.

The paper would be much more readable if it would start with a clear, formal problem formulation, followed by some schematic view on the overall flow and description on which parts are supervised, which parts are not. Also a tabular form and sample of the various kinds problems solved by this method could be listed in the beginning as a motivation with some clear description on how they fit the central paradigm and motivate the rest of the paper in a more concrete manner.

Instead, the paper is quite chaotic, switching between low-level and high level details, problem formulations and their solutions in a somewhat random, hard to parse order.

Both split and merge phases seem to make a lot of discrete choices in a hierarchical manner during training. The paper does not explain how those discrete choices are backpropagated through the network in an unbiased manner, if that is the case at all.

In general, the direction this paper is exciting, but the paper itself is a frustrating read in its present form. I have spent several hours on it without having to manage to achieve a clear mental image on how all the presented pieces fit together. I would revise my score if the paper would be improved greatly from a readability perspective, but I think it would require a major rewrite.

[Official Review · AnonReviewer1 · rating 4 · confidence 4 · 17 Dec 2016]
**Nice problem statement, nut too immature to publish**

The basic idea of this contribution is very nice and worth pursuing: how to use the powerful “divide and conquer” algorithm design strategy to learn better programs for tasks such as sorting or planar convex hull. However, the execution of this idea is not convincing and needs polishing before acceptance. As it is right now, the paper has a proof-of-concept feel that makes it great for a workshop contribution.

My main concern is that the method presented is currently not easily applicable to other tasks. Typically, demonstrations of program induction from input-output examples on well known tasks serves the purpose of proving, that a generic learning machine is able to solve some well known tasks, and will be useful on other tasks due to its generality. This contribution, however, presents a learning machine that is very hand-tailored to the two chosen tasks. The paper essentially demonstrates that with enough engineering (hardcoding the recurrency structure, designing problem-specific rules of supervision at lower recurrency levels) one can get a partially trainable sorter or convex hull solver.

I found the contribution relatively hard to understand. High level ideas are mixed with low-level tricks required to get the model to work and it is not clear either how the models operate, nor how much of them was actually learned, and how much was designed. The answer to the questions did hep, nut didn't make it into the paper. Mixing the descriptions of the tricks required to solve the two tasks makes things even more confusing. I believe that the paper would be much more accessible if instead of promising a general solution it clearly stated the challenges faced by the authors and the possible solutions.

Highlights:
+ Proof-of-concept of a partially-trainable implementation of the important “divide and conquer” paradigm
++ Explicit reasoning about complexity of induced programs
- The solution isn’t generic enough to be applicable to unknown problems - the networks require tricks specific to each problem
- The writing style pictures the method as very general, but falls back on very low level details specific to each task

[Official Review · AnonReviewer3 · rating 4 · confidence 2 · 20 Dec 2016]
**Promising idea but hard-to-reproduce in current state**

I was holding off on this review hoping to get the missing details from the code at

[Public Comment · ICLR 2017 conference · 16 Jan 2017]
**Author Response to third review?**

Dear authors,

do you plan to address the third reviewer's comments? Your responses could help bring some more clarity and improve the confidence for the final decision...

Thanks!

[Final Decision · Program Chairs · 06 Feb 2017]
**ICLR committee final decision**

The area chair agrees with the reviewers that this paper is not ready for ICLR yet. There are significant issues with the writing, making it difficult to follow the technical details. Writing aside, the technique seems somewhat limited in its applicability. The authors also promised an updated version, but this version was never delivered (latest version is from Nov 13).